# Zeb2 regulates differentiation of long-lived effector of invariant natural killer T cells

Tomonori Iyoda[1,5], Kanako Shimizu[1,2,5], Takaho Endo [3], Takashi Watanabe[3], Ichiro Taniuchi [4], Honoka Aoshima[1], Mikiko Satoh[1], Hiroshi Nakazato[1], Satoru Yamasaki[1] & Shin-ichiro Fujii [1,2✉]

After activation, some invariant natural killer T (iNKT) cells are differentiated into Klrg1[+] long-lived effector NKT1 cells. However, the regulation from the effector phase to the memory phase has not been elucidated. Zeb2 is a zinc finger E homeobox-binding transcription factor and is expressed in a variety of immune cells, but its function in iNKT cell differentiation remains also unknown. Here, we show that Zeb2 is dispensable for development of iNKT cells in the thymus and their maintenance in steady state peripheral tissues. After ligand stimulation, Zeb2 plays essential roles in the differentiation to and maintenance of Klrg1[+] Cx3cr1[+]GzmA[+] iNKT cell population derived from the NKT1 subset. Our results including single-cell-RNA-seq analysis indicate that Zeb2 regulates Klrg1[+] long-lived iNKT cell differentiation by preventing apoptosis. Collectively, this study reveals the crucial transcriptional regulation by Zeb2 in establishment of the memory iNKT phase through driving differentiation of Klrg1[+] Cx3cr1[+]GzmA[+] iNKT population.

[1] Laboratory for Immunotherapy, RIKEN Center for Integrative Medical Sciences (IMS), Yokohama, Kanagawa, Japan. [2] Program for Drug Discovery and Medical Technology Platforms, RIKEN, Yokohama, Kanagawa, Japan. [3] Laboratory for Integrative Genomics, RIKEN Center for Integrative Medical Sciences (IMS), Yokohama, Kanagawa, Japan. [4] Laboratory for Transcriptional Regulation, RIKEN Center for Integrative Medical Sciences (IMS), Yokohama, Kanagawa, Japan. [5] These authors contributed equally: Tomonori Iyoda, Kanako Shimizu. ✉email: shin-ichiro.fujii@riken.jp

nvariant Natural killer T (iNKT) cells are a unique type of innate T lymphocytes that stimulate or regulate subsequent immune responses. iNKT cells recognize glycolipids presented by the MHC class I-like molecule CD1d[1,2]. The rapid response of iNKT cells to their cognate antigens is characteristic of the innate immune response and allows the release of polarizing cytokines (IFN-γ and/or IL-4)[3]. These characteristics are shared with other effector lymphocytes, such as conventional effector or memory CD4+ and CD8+T cells, NK cells, mucosal-associated invariant T cells, and γδT cells[4,5]. In commitment to the iNKT cell lineage in the thymus, iNKT cell precursors are positively selected by double-positive thymocytes, which express CD1d-glycolipid complexes[6]. Because iNKT cells have divergent epigenetic expressions, transcription profiles, and gene expression programs, iNKT cells were separated into three subsets, NKT1, NKT2, and NKT17 cells, in which the key transcription factors, T-bet, Gata3, and Rorγt respectively, contribute to NKT1, NKT2, and NKT17 development[7–9]. Numerous genes were differentially expressed and regulated in each iNKT cell subset.

After egress from the thymus, these iNKT-cell subsets undergo further differentiation in the periphery. In C57BL/6 mice, individual iNKT cell subsets have a unique expression pattern of homing molecules[8], which may correlate with their difference in tissue distribution and function of the three NKT subsets[10]. In addition, several groups have analyzed the molecular details of iNKT cell differentiation by genome-wide analysis of gene expression, showing that most do not recirculate[8,11,12]. A recent study reported that the impact of tissue localization and homeostasis in peripheral organs rather than the thymus was determined by both iNKT subsets and activation status, which were proved by transcriptome and chromatin landscapes of iNKT cell studies[13]. Although iNKT cells express effector and/or memory T-cell markers, such as CD44, and CD69 in the steady state, how the phenotype and function of iNKT cells after activation can be controlled in peripheral tissues, remain to be determined.

Innate lymphocytes, which mount rapid effector responses, are thought to be short-lived. However, NK cells[14] and γδT cells[15] with memory type characteristics have previously been identified. We previously reported that Klrg1+ iNKT1 cells elicited after administration of α-GalCer-loaded DCs (DC/Gal) or -CD1d-expressing NIH3T3 cells are long-lived effector cells[16]. Klrg1+ iNKT cells, expressing the NKT1-like cytokine IFN-γ and cytotoxic molecules (FasL and granzyme A/B), persisted for several months in the lung and maintained long-term anti-tumor function. They displayed a robust secondary response to cognate antigen[16]. Analyses of CDR3β by RNA deep sequencing the iNKT cells in DC/Gal pre-injected mice demonstrated that some particular Klrg1+iNKT-cell clones accumulated, suggesting the selection of a certain TCR repertoire by an antigen. These findings demonstrated the effector memory-like Klrg1+ iNKT cells. These cells develop in a process dependent on the transient expression of Eomes peripherally[17]. We also demonstrated that Eomes plays an important role in the development of NKT1 cells in the thymus[17]; furthermore, other studies also characterized the Klrg1+ iNKT cells. Murray et al. also confirmed that α-GalCer treatment induced effector-like NKT cells in the spleen that expressed Klrg1 and Cx3cr1, which were derived from the NKT1 subset[13]. Plasit et al. also reported infiltration of Klrg1+ iNKT cells at the tumor site and draining lymph nodes elicited by intratumoral injection of CpG and α-GalCer, which enhanced the abscopal effect in the antitumor effect of CD8 T cells[18]. However, it remains unknown which specific molecules play a key role in the differentiation of Klrg1+ iNKT cells in the periphery without affecting the thymus.

Zinc finger E-box-binding homeobox 2 (Zeb2) is a transcription factor that acts as a transcriptional repressor by cooperating with activated SMAD proteins and by recruiting of either the corepressor C-terminal binding protein or histone deacetylase complexes, such as nucleosome remodeling and deacetylase[19]. Initially, Zeb2 mediates the epithelial-to-mesenchymal transition through repression of epithelial genes. Recently, the role of Zeb2 in the differentiation and function of NK and T cells, as well as myeloid cells, DCs, and macrophages, was revealed[20–27]. However, the importance of Zeb2 in iNKT cells has not yet been reported; but Zeb2 has been reported to cooperate with Klrg1 in T and NK cells. In this study, we examined whether Zeb2 is involved in the development of iNKT cells in the thymus or periphery using T-cell-specific Zeb2-conditional knockout mice (Zeb2-cKO).

## Results

**Klrg1+ iNKT cells elicited by DC/Gal express Zeb2.** Klrg1+ iNKT cells increased after the administration of DC/Gal[16]. We analyzed the pattern of gene expression, especially transcription factors of lung iNKT cells from naïve and DC/Gal-immunized mice using RNAseq data previously reported[17] (Supplementary Fig. 1a). Fold change (DC/Gal versus naïve) showed the highest upregulation of Zeb2 (Fig. 1a). After sorting Klrg1+ iNKT cells on day 7, we confirmed that Zeb2 transcripts were highly expressed in Klrg1+ iNKT cells (Fig. 1b). In T and NK cells, it was previously shown that Zeb2 expression is dependent on T-bet in both Th1 and NK cells[20–22]. To determine whether the T-bet also functions in iNKT cells for Zeb2 induction, when T-bet KO mice were administered with DC/Gal, a lower frequency of Klrg1+ iNKT cells was detected (Supplementary Fig. 1b–d). Furthermore, these Klrg1+ iNKT cells from T-bet KO mice immunized with DC/Gal did not express Zeb2 (Fig. 1b *right*), indicating that T-bet is essential for Zeb2 induction in activated iNKT cells. To investigate the potential role of Zeb2 in iNKT-cell activation, we generated CD4-Cre Zeb2f/f (Zeb2-cKO) mice that lack Zeb2 activity specifically in T cells. The frequency of conventional CD4+ and CD8+ T cells in the spleen of Zeb2-cKO mice was normal (Supplementary Fig. 2a). The frequency of iNKT cells and the expression intensity of invariant TCR expression in iNKT cells in spleen were similar among wild-type mice, CD4-Cre negative Zeb2f/f littermate mice, and Zeb2-cKO mice (Fig. 1c). In addition, we analyzed whether steady-state iNKT cells develop normally in the lungs of Zeb2-cKO mice. We failed to detect any significant differences in the number and cytokine expression of steady-state lung iNKT cells in both WT and Zeb2-cKO mice (Supplementary Fig. 2b, c). Next, we examined the generation of Klrg1+ iNKT cells. The frequency and cell number of Klrg1+ iNKT cells in the lung and spleen of WT mice was increased a week after administration of DC/Gal (Fig. 1d–f). In contrast, the frequency of Klrg1+ iNKT cells in Zeb2-cKO mice was significantly lower than in WT mice and in mice with a floxed *Zeb2* allele lacking the Cre transgene (littermate controls) (Fig. 1d–f). We previously reported that in DC/Gal-immunized mice, the expression of *Tbx21* in Klrg1+ iNKT cells is considerably higher than that in Klrg1− iNKT cells[17]. We subsequently compared the expression of *Tbx21* in Klrg1+ or Klrg1− iNKT cells between DC/Gal-injected Zeb2-cKO and WT mice. (Fig. 1g). Our findings revealed that the expression of *Tbx21* in the Klrg1+ iNKT cells of DC/Gal-injected WT mice was higher than that in DC/Gal-injected Zeb2-cKO mice. In contrast, the expression of *Tbx21* in the Klrg1+ cells of DC/Gal-injected Zeb2-cKO mice was similar to that in the Klrg1− iNKT cells of DC/Gal -injected WT and Zeb2-cKO mice (Fig. 1g). In summary, the frequency of Klrg1+ iNKT cells was considerably lower in Zeb2-cKO mice after administration with DC/Gal (Fig. 1e). Additionally, the expression of T-bet in the

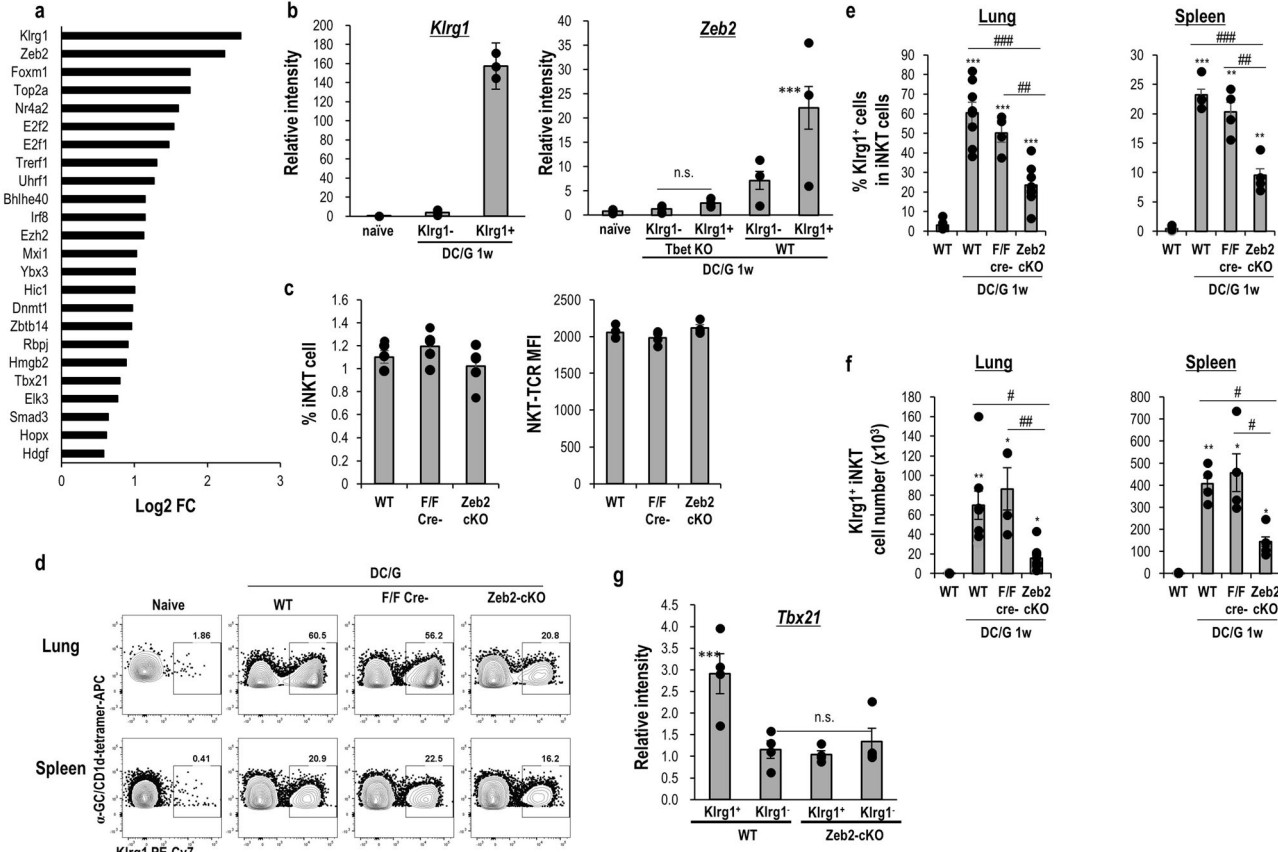

**Fig. 1 Zeb2 expression in Klrg1+ iNKT cells. a** Expression of transcription factors (TFs) in lung iNKT cells. Lung iNKT cells from DC/Gal-immunized mice on day 7 and lung iNKT cells from naïve mice were sorted and analyzed by RNAseq (accession code GSE128069). Log2 Fold change (Log2FC) of each TF (DC/Gal FPKM/naïve FPKM) are shown ($n = 3$/group). **b** Klrg1 and Zeb2 expression of the indicated iNKT cells. Lung Klrg1+ and Klrg1− iNKT cells from DC/Gal-immunized mice on day 7 and lung iNKT cells from naïve mice were sorted. TF expression was analyzed by qPCR. ($n = 3$/group, mean ± SEM, ***$p < 0.001$ (WT Klrg1+ iNKT vs others), Tukey test). **c** Percentage of CD1d-tet+TCRβ+iNKT cells in WT, littermate, and Zeb2-cKO spleens. In the right panels, the TCR expression intensity of the spleen is shown. ($n = 5$, mean ± SEM, left panels) **d** Induction of Klrg1+ iNKT cells by DC/Gal immunization. The WT, littermate, and Zeb2-cKO mice were immunized with DC/Gal. One week later, the lung and spleen iNKT cells were analyzed by flow cytometry. The expression Klrg1 in iNKT cells from naïve WT or DC/Gal-immunized WT, littermate (F/F Cre-), or Zeb2-cKO mice. The frequency (**e**) and absolute cell number (**f**) of Klrg1+ cells in iNKT cells in the indicated group ($n = 4$–$8$, mean ± SEM). *$p < 0.05$, **$p < 0.01$, ***$p < 0.001$ Student's $t$ test to WT naïve, #$p < 0.05$, ##$p < 0.01$, ###$p < 0.001$ ANOVA Tukey–Kramer method in the immunized group. **g** Tbx21 expression of the indicated iNKT cells. Lung Klrg1+ and Klrg1− iNKT cells from DC/Gal-immunized WT and Zeb2-cKO mice were sorted on day 7. The expression was analyzed by qPCR. ($n = 4$/group, mean ± SEM ***$p < 0.001$ (WT Klrg1+ iNKT vs others), Tukey test).

Klrg1+ iNKT cells in DC/Gal-injected Zeb2-cKO mice was much lower than that of Klrg1+ iNKT cells in DC/Gal-injected WT mice (Fig. 1g).

**Zeb2 is required for the differentiation of Klrg1+Cx3cr1+GzmA+ iNKT cells.** Previously, we demonstrated that Klrg1+ iNKT cells express granzyme A (GzmA) exclusively[16]. Therefore, we initially confirmed high expression of Gzma transcripts in sorted Klrg1+ iNKT cells, but not Klrg1− iNKT cells using real-time PCR from the lungs after administration of DC/Gal (Fig. 2a). Murray et al. reported that Cx3cr1 is also a marker for the antigen-exposed effector, NKT1 type Klrg1+ iNKT cells[13]. Therefore, we assessed GzmA and Cx3cr1 expression levels in Klrg1+ iNKT cells by flow cytometry. Most Klrg1+ iNKT cells expressed GzmA and Cx3cr1 in the lung, but were also detected in the spleen at a lower frequency (Fig. 2b, c). While the frequency of Cx3cr1+ GzmA + in Klrg1+ iNKT cells in WT and littermate control mice on day7 after DC/Gal administration was detected at approximately 40–50%, these populations were mostly diminished in Zeb2-cKO mice (Fig. 2d).

In Zeb2-deficient conventional T cells, Klrg1loCD127hi memory precursor cells increased during viral infection[20,21]. However, expression pattern of Klrg1 and CD127 has not been analyzed in iNKT cells. Klrg1+ iNKT cells downregulated expression of CD127 in WT and Zeb2-cKO mice immunized with DC/Gal (Fig. 2e). We next examined whether Klrg1+ iNKT cells expressed markers of T-cell exhaustion. Klrg1+ iNKT cells did not exhibit higher expression of PD-1, CTLA4, TIM3, and Lag3 (Fig. 2f). Therefore, the data indicate that Klrg1+ iNKT cells are not likely to be exhausted cells.

It has been shown iNKT_FH are generated after an administration with α-GalCer[28,29]. Consistent with previous reports, our experimental setting also generated PD-1+Cxcr5+ iNKT_FH cells in WT as well as Zeb2-cKO mice (Fig. 2g). These results suggest Zeb2 plays a selective role in driving differentiation of Klrg1+ Cx3cr1+GzmA+ iNKT cells.

**Normal development of iNKT-cell subsets in the thymus in Zeb2-deficient mice.** As shown in Supplementary Fig. 2b, the absolute cell numbers of iNKT cells in the thymus of WT and Zeb2-cKO mice were comparable. Next, we carefully assessed

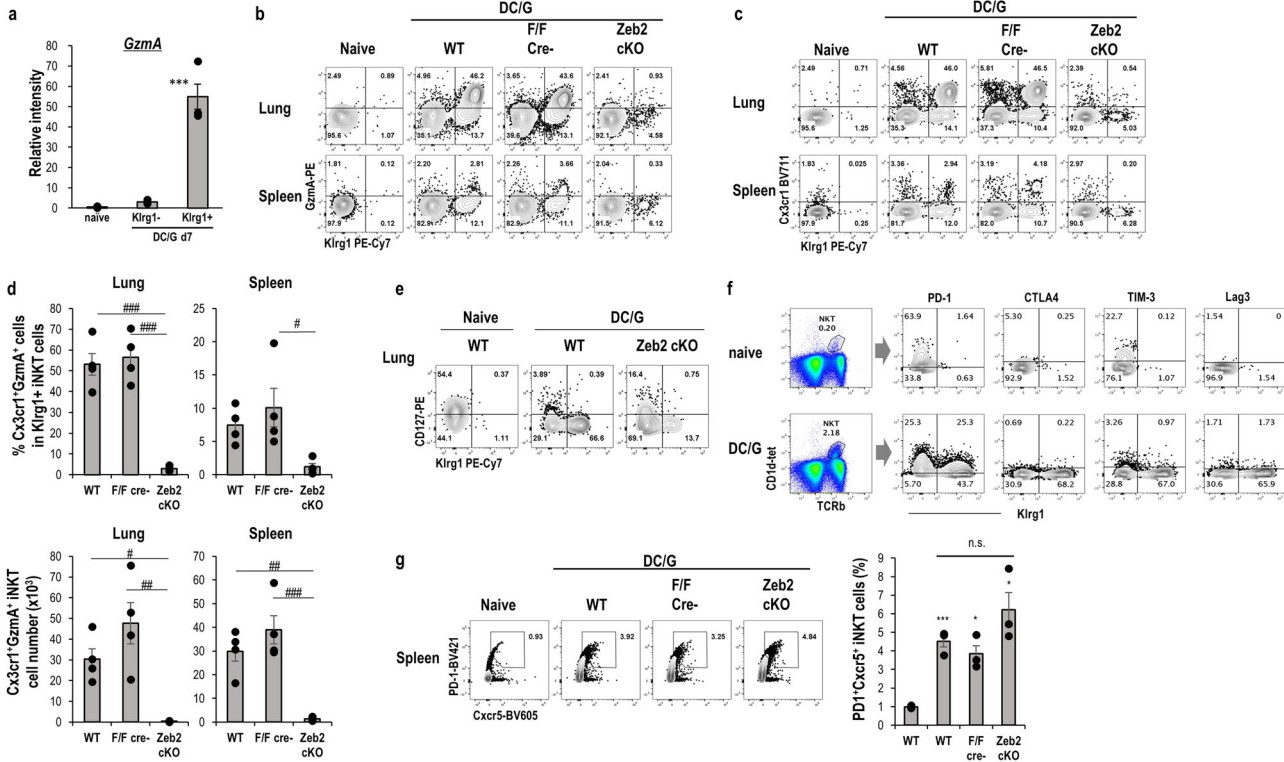

**Fig. 2 Characterization of Klrg1⁺ iNKT cells. a** *GzmA* expression in iNKT cells. As shown in Fig. 1b, *GzmA* expression in the indicated iNKT cells was analyzed using qPCR. ($n = 3$/group, mean ± SEM ***$p < 0.001$ (WT Klrg1⁺ iNKT vs others), Tukey test). Phenotype of Klrg1⁺ iNKT cells. WT, littermate, or Zeb2-cKO mice were immunized with DC/Gal, and the expression of GzmA (**b**) and Cx3cr1 (**c**) of Klrg1⁺ iNKT cells in the lung and spleen was analyzed by flow cytometry. **d** The frequency (upper) and cell number (lower) of GzmA⁺Cx3cr1⁺ in Klrg1⁺ iNKT cells were summarized. ($n = 4$, mean ± SEM). #$p < 0.05$, ##$p < 0.01$, ###$p < 0.001$ ANOVA Tukey–Kramer method. **e** The expression of CD127 of Klrg1⁺ iNKT cells in the lung from WT and Zeb2 cKO immunized with DC/Gal ($n = 4$, mean ± SEM). **f** The inhibitory molecules of Klrg1⁺ iNKT cells in the lung from WT immunized with DC/Gal. **g** Induction of NKT_FH. Representative dot plot of NKT_FH markers (PD-1 and CXCR5) in iNKT cells (left) and frequency (right) of NKT_FH in the spleen ($n = 4$, mean ± SEM).*$p < 0.05$, **$p < 0.01$, ***$p < 0.001$ Student's *t* test to WT naïve.

whether Zeb2 plays a role in the development of iNKT cells in the thymus. Closer analysis of the thymus revealed that the frequency and absolute number of CD24⁺CD44⁻ stage 0, CD24⁻CD44⁻NK1.1⁻ stage 1, CD44⁺NK1.1⁻ stage 2, and CD44⁺NK1.1⁺ stage 3 were similar in WT mice, littermate control, and Zeb2-cKO mice (Fig. 3a, b). We analyzed the expression of transcription factors related to iNKT-cell differentiation, such as T-bet, Gata3, Rorγt, of NKT1, NKT2 and NKT17 subset respectively, and the master regulator (PLZF) of all iNKT cells. The development of iNKT-cell subsets was almost the same among WT, littermate control, and Zeb2-cKO mice (Fig. 3c–e). Therefore, these results suggest Zeb2 does not affect the development of iNKT-cell subsets in the thymus. In fact, when the expression of *Zeb2* in each stage of iNKT cells in the thymus is examined, *Tbx21* expression is upregulated as the stage progresses, whereas *Zeb2* expression is barely observed (Fig. 3f).

Two recent papers separated NKT1 subset into three subsets of NKT1 in steady state of the thymus and peripheral tissues: C0 iNKT cell[30] (iNKT1a[31]), C1 iNKT cell(2B4⁻)[30] (iNKT1b[31]) and C2 iNKT cell(2B4⁺)[30] (iNKT1c[31]). In the current study, we verified the NKT1 subsets in WT and Zeb2-cKO mice similarly (Supplementary Fig. 2d, e). Furthermore, having sorted C1 iNKT and C2 iNKT cells from the thymus of WT mice, each type was separately transferred to Rag⁻/⁻ mice (Supplementary Fig. 2f), and on day 5 after the transfer, DC/Gal were administered. Subsequent analysis of Klrg1⁺ iNKT cells in the lungs 1 week later revealed the expression of Klrg1 in the C1 iNKT and C2 iNKT cells-transferred mice (Supplementary Fig. 2g), thereby

indicating that both C1 and C2 iNKT cell subsets can undergo differentiation to Klrg1⁺ iNKT cells.

**Zeb2 maintains long-lived Klrg1⁺ iNKT effector cells.** Because Klrg1⁺ iNKT cells in the lung were maintained for a long period as effector memory several months after activation[16], we next examined the role of Zeb2 in the longevity of effector iNKT cells 4 weeks after DC/Gal administration. The frequency of iNKT cells in the lungs in Zeb2-cKO mice were almost similar to those of WT mice (Supplementary Fig. 3a). Klrg1⁺ iNKT cells in the lungs of DC/Gal-immunized WT mice and littermate controls were maintained with relatively high frequency compared to the spleens of DC/Gal-immunized WT mice and littermate controls (Fig. 4a, b). One month after immunization the Klrg1⁺ iNKT cells exhibited a CD27loCD43hiCD62L⁻CD69⁻CD122⁺ NK1.1⁺NKG2DhiLy6ChiCD49d⁺ phenotype, which are different from iNKT cells in the steady state (Supplementary Fig. 3b). A lower frequency of Klrg1⁺ iNKT cells was detected in Zeb2-cKO mice (Fig. 4a, b), and then Klrg1⁺ Cx3cr1⁺GzmA⁺ iNKT cells were greatly diminished (Fig. 4c, d). These findings thus indicate that Zeb2 plays a role in the differentiation of NKT1 cells from activation to the maintenance of long-term effector Klrg1⁺ Cx3cr1⁺ GzmA⁺ iNKT cells in peripheral tissues.

**The generation of Klrg1⁺ long-lived iNKT effector cells was independent of the thymus.** We suspected whether newly recruited iNKT cells from the thymus were continuously

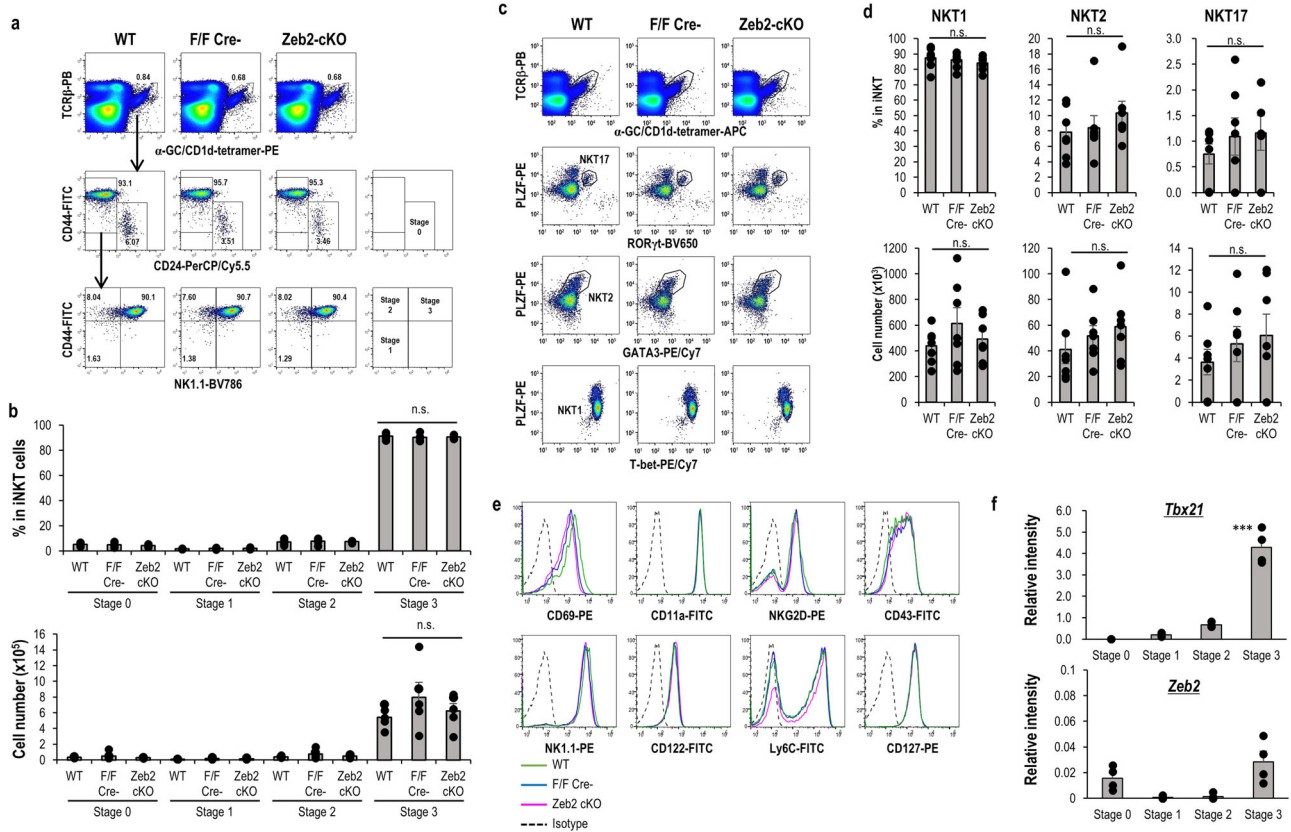

**Fig. 3 Development of iNKT cells in the thymus in Zeb2-cKO. a** Percentage of CD1d-tet+TCRβ+iNKT cells (upper) and CD24, NK1.1, and CD44 staining in gated iNKT cells (middle and lower) in WT, littermate, and Zeb2-cKO mice. **b** Percentage (upper) and cell number (lower) of stage 0 (CD24+CD44−NK1.1−CD69+), stage 1 (CD24−CD44−NK1.1−), stage 2 (CD24−CD44+NK1.1−), and stage 3 (CD24−CD44+NK1.1+) cells in the indicated mice. (n = 5, mean ± SEM). **c** Representative flow cytometry data showing the expression of PLZF, Rorγt, Gata3, and T-bet by thymic iNKT in WT, littermate, and Zeb2-cKO mice. **d** Percentage (upper) and cell number (lower) of NKT1 (PLZFdimT-bet+), NKT2 (PLZFhighGata3+), and NKT17 (PLZF+Rorγt+) subsets in the thymus of WT, littermate, and Zeb2-cKO mice (n = 4–6, mean ± SEM). **e** Representative histograms showing the expression of CD69, CD11a, NKG2D, CD43, NK1.1, CD122, Ly-6C, and CD127 in thymic iNKT cells from WT (green), littermate (blue), and Zeb2-cKO (magenta) mice. Similar data was obtained from at least four independent experiments. **f** The expression of *Tbx21* (upper) and *Zeb2* (lower) of iNKT cells in stage 0, 1, 2 and 3. (n = 4, mean ± SEM) Stage 3 vs others in Tbx21 ***p < 0.001 ANOVA Tukey–Kramer method.

activated or that iNKT effector cells were differentiated to a long-lived form in peripheral tissues independent of the thymus. Thymectomized mice or sham-treated mice were administered DC/Gal two weeks later. Four more weeks later, the frequency and number of iNKT cells were evaluated in the lungs and spleen (Fig. 5 and Supplementary Fig. 4). The frequency and cell number of total iNKT cells in the lung were decreased in DC/Gal-administered thymectomized mice compared with DC/Gal-injected sham mice (Fig. 5a and Supplementary Fig. 4a). However, the percentages and cell numbers of Klrg1+ iNKT cells, particularly Cx3cr1+ GzmA+ effector iNKT cells, were comparable. (Fig. 5b–e and Supplementary Fig. 4b). These findings suggest that Klrg1+ iNKT cells were not replenished from the thymus, but were maintained in the peripheral tissue.

**Single-cell transcriptome analysis of Klrg1+ iNKT cells from WT and Zeb2-cKO mice.** The question arose as to why Klrg1+ iNKT appeared transiently at a low frequency in Zeb2-cKO mice after activation, and disappeared later. In order to elucidate the heterogeneity and differentiation trajectory of lung Klrg1+ iNKT cell population, we sorted lung Klrg1+ iNKT cells one week after immunization with DC/Gal, and analyzed them using single-cell transcriptome analysis. After integrating iNKT cells from littermate control and Zeb2-cKO mice, the data were dimensionally reduced by UMAP with unbiased clustering (Supplementary

Fig. 5). In fact, UMAP plot showed unbiased total of 12 clusters (Supplementary Fig. 5). To identify the cell state of each cluster, mean read counts and the ratio of mitochondrial genes were calculated. After removing dead cells, doublets, or multiplets in addition to a non-NKT cluster, we found a total of eight clusters of iNKT cells (Fig. 6a). Next, the cluster frequency was analyzed (Fig. 6b). Clusters C1, 9, 10, and 11 were found only in littermate control, whereas C2 was found only in Zeb2-cKO; C3 and C4 had a higher ratio in KO than in littermate control; and C7 was comparable. This suggests that C2, 3, 4 and 7 were independent of Zeb2. We analyzed gene signatures in each cluster of Klrg1+ iNKT cells and represented themt as a heatmap (Fig. 6c). C4 is *Gata3*hi *Tbx21*lo *IL17rb*hi, typical of the NKT2 subset. C3 was unclassified because it expressed mixed gene signatures (*Tbx21*lo *Lef1*hi *Plac8*hi *Gata3*hi *Fhl2*hi *CXCR6*hi) of immature NKT1 and NKT2 subsets. We found that Klrg1+ iNKT cells on day 7 included some NKT2 (C4) and unclassified cells (C3) in a small population. Given that in both WT and Zeb2-cKO mice, NKT2 cells were clearly detected in C4, this indicates that the differentiation of NKT2 to Klrg1+ iNKT cells is not dependent upon Zeb2. Similarly, the unclassified Klrg1+ cells of C3 were identified as Zeb2-independent cells. By contrast, the main cluster C2 of Zeb2-cKO Klrg1+ iNKT expressed a low RNA content and low expression of mitochondrial-related genes, suggesting a dying population (Supplementary Fig. 5).

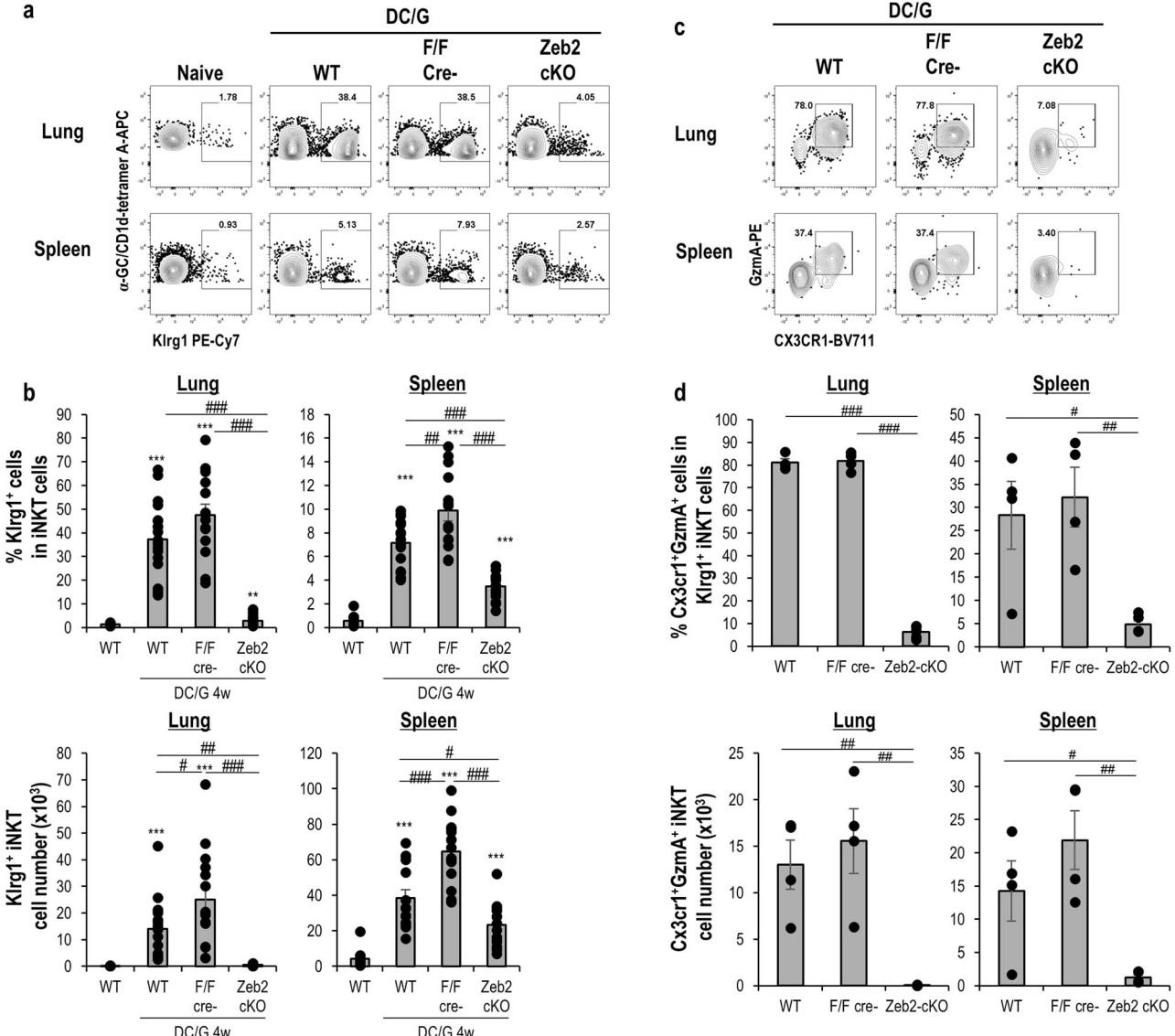

**Fig. 4 Maintenance of Klrg1+ iNKT cells.** The WT, littermate, and Zeb2-cKO mice were immunized with DC/Gal. Four weeks later, the lung, and spleen iNKT cells were analyzed using flow cytometry. **a** Representative flow cytometry data showing the expression Klrg1 in iNKT cells in the lung (upper) and spleen (lower) from naïve WT or DC/Gal-immunized WT, littermate, or Zeb2-cKO mice. **b** Frequency (upper) and cell number (lower) of Klrg1+ cells in iNKT cells in the indicated group ($n = 12$–19, mean ± SEM). **c** Cx3cr1+ GzmA+ cells gating on Klrg1+ iNKT cells s in the lung (upper) and spleen (lower) from the indicated group. The data represents four independent experiments. **d** Frequency (upper) and cell number (lower) of Cx3cr1+granzyme A+Klrg1+ iNKT cells in the indicated group. ($n = 4$–5, mean ± SEM). *$p < 0.05$, **$p < 0.01$, ***$p < 0.001$ Student's $t$ test to WT naïve, #$p < 0.05$, ##$p < 0.01$, ###$p < 0.001$ ANOVA Tukey–Kramer method in the immunized group.

In contrast, C1, C7, C9, C10, and C11 were in the NKT1 lineage. We verified that many iNKT cells among NKT1 clusters expressed *Zeb2* as well as *Klrg1* (Supplementary Fig. 6). The trajectory analysis using psylingshot showed that the differentiation goes in the following order, i.e., C7 → C11 → C1 → C9 → C10 (Fig. 6d). *Zeb2* expression was relatively low at C7 as the starting point, peaked at C9, and decreased at C10 (Fig. 6e). Patterns of differentially expressed genes between these clusters in NKT1 lineages supported this trajectory pattern. For instance, C7 exhibited *CD5*hi, which reflects TCR signal, and high expression of NF-κB pathway genes and C9 showed a high level of gene expression of cytotoxic molecules and chemokines like NK cells (Fig. 6f). In Klrg1+ iNKT cells from Zeb2-cKO mice, C11, C1, C9 and C10 were almost lost, suggesting the differentiation is inhibited from C7 to C11 in the absence of Zeb2. Instead, C7 skewed to C2 showing dying signatures. Long-lived Klrg1+

Cx3cr1+GzmA+ iNKT cells originated from C11, C1, and C9. Furthermore, besides *GzmA, Cx3cr1, Ccl5, Klf2, Klf3*, and *GzmB* also behaves similarly with *Zeb2* (Fig. 6e). Therefore, we clarified the branch point of iNKT-cell differentiation where Zeb2 functions to drive differentiation from C7 to C11 and protect precursors from apoptotic pathway. Thus, we showed that whereas Zeb2 plays a role in the differentiation of NKT1 cell into the memory phase, it does not influence the differentiation of NKT2 subset cells.

**Zeb2 deficient Klrg1+ NKT cells underwent apoptosis.** We checked the protein level to determine whether Klrg1+ iNKT cells in DC/Gal-injected Zeb2-cKO mice induced apoptosis (Fig. 7). When the cell proliferation marker Ki-67 and apoptosis Annexin-V were examined in DC/Gal-injected WT and Zeb2-cKO mice 7 days

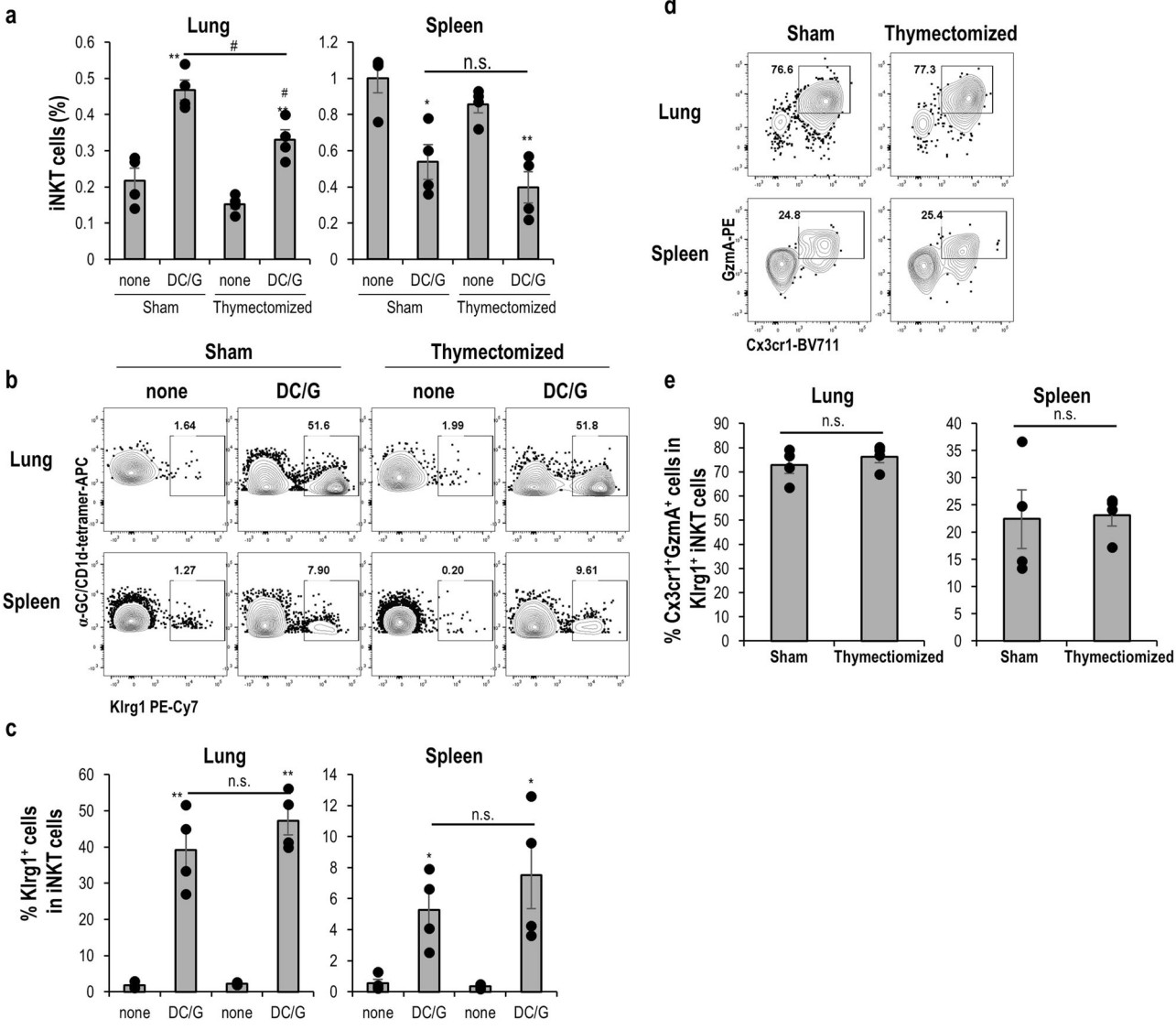

**Fig. 5 Maintenance of long-lived iNKT cells in a thymus-independent manner.** Two weeks after thymectomy or sham treatment, the mice were immunized with DC/Gal. Four weeks later, lung and spleen iNKT cells were analyzed by flow cytometry. All thymectomized mice were evaluated for complete removal of the thymus during tissue collection. **a** Frequency of iNKT cells in the indicated group ($n = 4$, mean ± SEM). **b** The expression Klrg1 in iNKT cells from naïve WT or DC/Gal-immunized thymectomized or sham mice. **c** Frequency of Klrg1+ cells in iNKT cells in the indicated group ($n = 4$, mean ± SEM). **d**, **e** Expression and frequency of Cx3cr1+GzmA + cells gating on Klrg1+ iNKT cells in the indicated organs ($n = 4$, mean ± SEM). The data represents four independent experiments. *$p < 0.05$, **$p < 0.01$, ***$p < 0.001$ Student's $t$ test between non-treated group and DC/Gal treated group in sham or thymectomized mice, #$p < 0.05$ Student's $t$ test between sham and thymectomized mice in DC/Gal treated group.

after immunization, the expression of Ki-67 in the Klrg1+ iNKT cells was lower (Fig. 7a) and more Annexin-V-positive in the Klrg1+ iNKT cells were present in DC/Gal-injected Zeb2-cKO mice than those in the DC/Gal-immunized WT mice (Fig. 7b), indicating that Klrg1+ iNKT cells are apoptotic and cannot be maintained when Zeb2 is deficient. In contrast, however, we did not detect any significant differences in Ki-67+ T cells or Annexin-V-positive T cells in the TCRβ+tetramer− T cells between WT and Zeb2-cKO mice (Fig. 7a, b). Together, consistent with the single-cell transcriptome results, Klrg1+ iNKT cells were apoptotic and unmaintainable when Zeb2 was deficient (Fig. 7c).

**Secondary boosting effect in Zeb2-cKO mice.** Finally, we also sought to determine whether an antigen-specific secondary response could be maintained in DC/Gal-vaccinated Zeb2-cKO mice (Fig. 8). Mice were rechallenged with DC/Gal 7 months after

the first vaccination and analyzed 1 week later. The total number of Klrg1+ iNKT cells in WT mice after boosting with DC/Gal was almost three times higher than in WT mice with DC/Gal immunization, consistent with a previous report[16]. The number of Klrg1+ iNKT cells in vaccinated Zeb2-cKO mice did not increase after the boost with DC/Gal, but was like the primary response. The recall response, which indicates the rapid response of pre-existing memory cells by re-exposure to the same antigen, was maintained in WT mice, but not in Zeb2-cKO mice. Therefore, Zeb2 may play an important role in the memory formation of Klrg1+ iNKT cells.

## Discussion

In this study, we determined the role of Zeb2 in antigen-driven differentiation of iNKT cells in peripheral tissues. Zeb2 did not affect the differentiation of NKT1, NKT2, or NKT17 cells in the thymus, but significantly affected the generation of long-lived effector Klrg1+

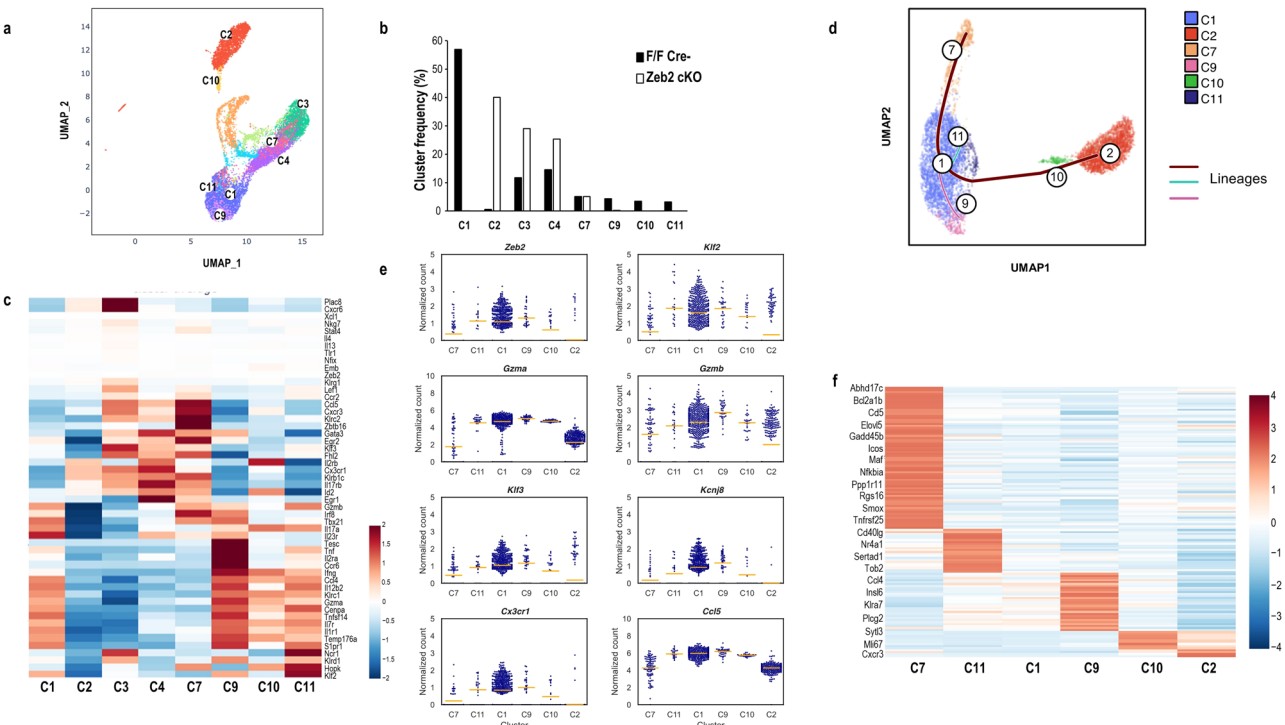

**Fig. 6 Analysis of the single-cell transcriptome of Klrg1$^+$ iNKT cells from DC/Gal-immunized WT and Zeb2-cKO mice.** Lung Klrg1$^+$ iNKT cells were sorted from DC/Gal-immunized WT and Zeb2-ckO mice and analyzed using sc-RNAseq. **a** Integrated UMAP of Klrg1$^+$ iNKT cells derived from WT and Zeb2-ckO mice. Eight clusters of iNKT cells were identified. **b** Frequency of clusters. Black and white bars indicate WT and Zeb2-ckO mice, respectively. **c** Heatmap of gene signatures in each cluster of Klrg1$^+$ iNKT cells. **d** Trajectory analysis of iNKT1 lineage. **e** Gene expression in iNKT1 lineage clusters with similar pattern of *Zeb2*. **f** Heatmap of DEG in iNKT1 lineage clusters.

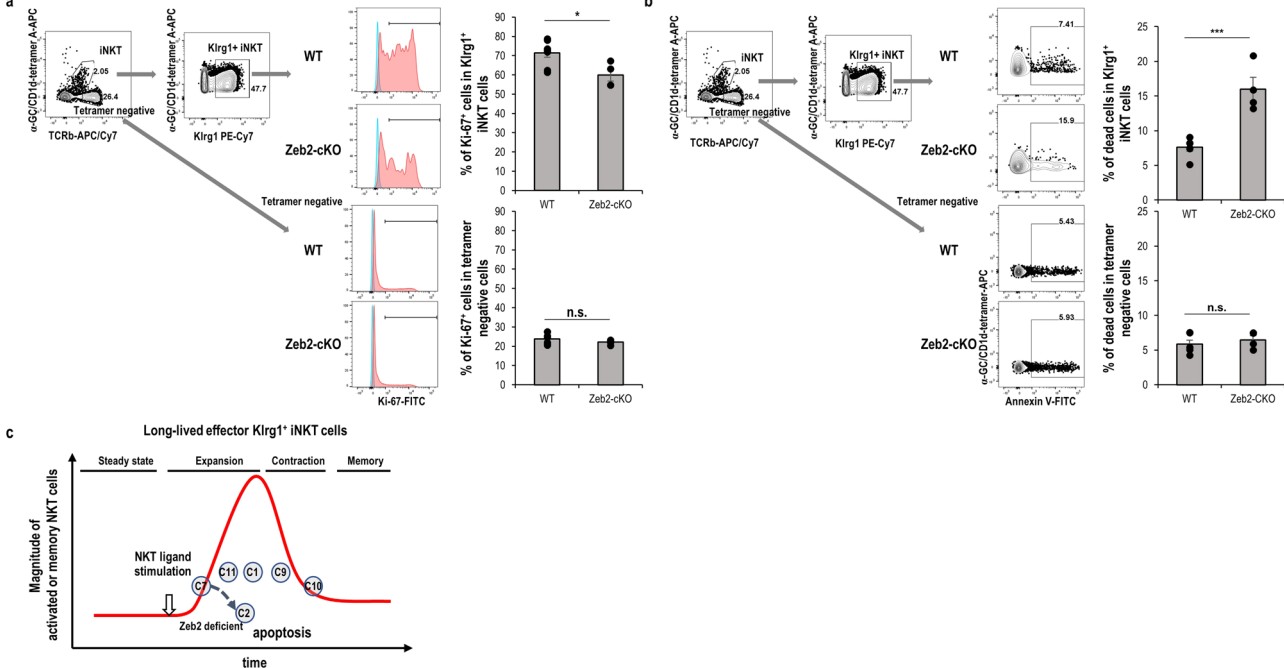

**Fig. 7 Apoptosis and proliferation of Klrg1$^+$ iNKT cells from DC/Gal-immunized WT and Zeb2-cKO mice.** Klrg1$^+$ iNKT cells cells or TCRβ$^+$tetramer$^-$ T cells in lung for Ki-67 (**a**) and Annexin-V (**b**) were evaluated one week after DC/Gal injection. The flow data is representative histograms (isotype; blue, Ki-67; red). The bar graph in (**a**) summarizes three independent experiments (WT, $n = 8$; Zeb2-cKO, $n = 4$; mean ± SEM). The bar graph in (**b**) summarizes four independent experiments (WT, $n = 6$; Zeb2-cKO, $n = 4$; mean ± SEM). *$p < 0.05$, ***$p < 0.001$, Student's $t$ test. **c** Catoon of differentiation of long-lived effector iNKT cells.

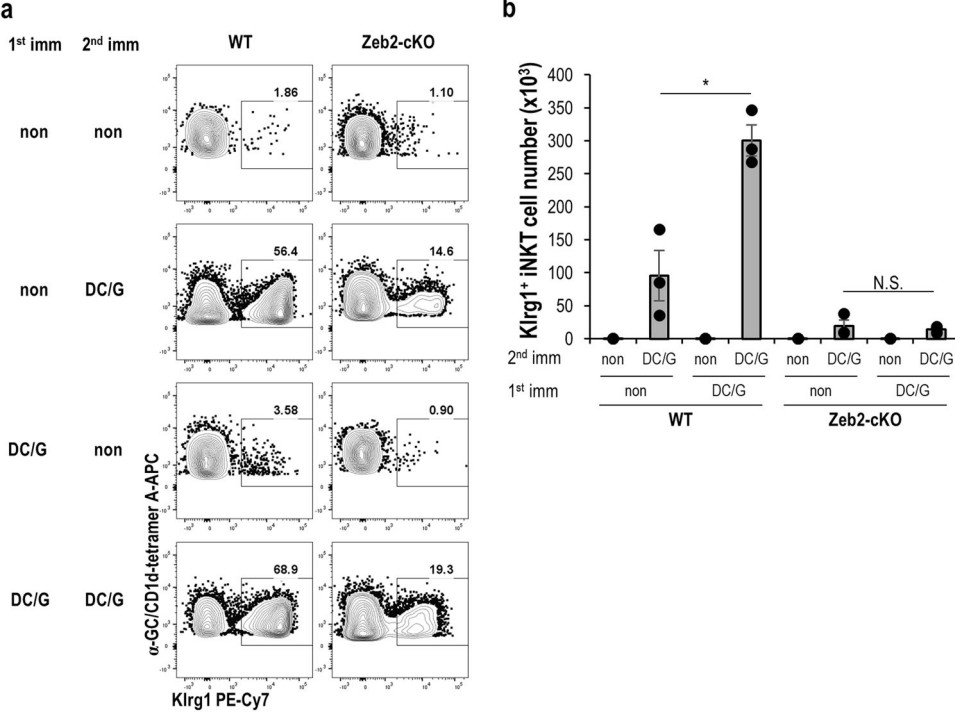

**Fig. 8 Zeb2-deficient iNKT cells fail to establish long-term memory.** WT and Zeb2-cKO mice were immunized with or without DC/Gal as the first immunization ($n = 6$). Seven months later, half of the mice were administered DC/Gal as the second immunization. **a** Dot plot data represent three independent experiments. **b** The number of Klrg1+ iNKT cells in the lungs was measured seven days after the second immunization ($n = 3$, mean ± SEM). *$p < 0.05$, Student's $t$ test.

iNKT cells from the NKT1 subset in the lungs after antigen stimulation. In our previous reports on mice immunized with DC/Gal, Klrg1+ iNKT cells were characterized by the expression of several markers, including the CD43hiCD49dhiNKG2Dhi GzmA + phenotype, Ccl3, Ccl4, IFN-γ, and Tbx21[16]. Using RNAseq and ATAC-seq, Murray et al. confirmed that iNKT effector cells showed Klrg1+ GzmA + Cx3cr1+IFN-γ+[13]. Both findings suggest that they are derived from the NKT1 subset. In this study, we also verified that Klrg1+ prolonged effector iNKT cells were obtained from the NKT1 subset in a Zeb2-dependent manner. Regarding the steady state of iNKT cells, two recent studies demonstrated that naïve NKT1 cells were separated into three subsets in the thymus and peripheral tissues: C0 (iNKT1a), C1 (iNKT1b), and C2 (iNKT1c)[30,31]. The C2 (iNKT1c) subset shows higher levels of cytotoxic molecules and IFN-γ expression. These three NKT1 subsets were maintained in the steady state in the presence of IL-15. In contrast to cytokine signaling, TCR stimulation with DC/Gal can differentiate naïve NKT1 cells to long-lived iNKT cells. We verified that both C1 iNKT1 and C2 iNKT1 subsets from thymus differentiated into Klrg1+ iNKT cells in the periphery, i.e., lung after administration with DC/Gal.

The persistent expression of Klrg1, which is enforced by T-bet, Zeb2, and Id2, has been associated with senescence and terminal differentiation[20,21,32]. During acute infection, effector CD8+ T cells are divided into two populations: the Klrg1−CD127hi (MPEC) population is supposed to form a long-lived memory pool, whereas the Klrg1+CD127low (SLEC) population comprises short-lived effector cells[33,34]. In contrast, some Klrg1+CD127low CD8+ T cell population persists in the memory phase and exhibits robust protective functions during rechallenge with bacteria or viruses[35,36]. These CD8+ T cells are termed long-lived effector cells (LLEC). LLEC, which is a circulating memory population, expresses Klrg1, GzmA and Cx3cr1; this profile resembles the Klrg1+ iNKT cells.

Although Zeb2 is involved in the regulation of NK cells and T cells, there are similarities and differences in their activation signals and the maintenance of memory function. First, Zeb2 is required for NK cell differentiation. Zeb2-deficient NK cells were found to have impaired differentiation into the CD27−CD11b+terminal maturation stage due to NK apoptosis[22]. By contrast, Zeb2 had no specific effect on T-cell differentiation in the steady state. However, in LCMV infection models, Zeb2-deficient CD8+ T cells show a marked decrease in antigen-specific CD8+ T cells in the effector phase[20,21]. The generation of Klrg1(hi) effector memory T-cell populations was significantly altered, and Zeb2-deficient memory CD8+ T cells were reduced in secondary reactions[20]. In the case of iNKT cells, like T cells, Zeb2-deficient iNKT cells can develop normally in the thymus, at a normal frequency. However, apoptosis of some of these cells was revealed during differentiation after activation, similar to that of NK cells.

Several studies suggest that T-bet, a transcription factor, may regulate Zeb2 expression in T cells and NK cells[20–22]; furthermore, Zeb2 expression is implied in the importance of c-Myb[37]. T-bet binds to the Zeb2 locus to induce Zeb2 expression and both transcription factors appear to be required for optimal binding to the T-bet-regulated cis-regulatory region. By contrast, c-Myb-mediated repression of Zeb2 may involve interaction with the coactivator p300 via the induction of repressive non-coding RNA[37]. Because we did not find an enhancement in Zeb2 expression in Klrg1+ iNKT in T-bet KO mice, Zeb2 expression must be regulated by T-bet in the NKT1 subset, as well as in T cells and NK cells. The antagonistic relationship between the actions of the transcription factors Zeb1 and Zeb2, and micro RNA (miR)-200 regulates and determines the memory capacity of effector T cells. In this circuit, Zeb1 with miR-200 family microRNAs repress Zeb2 expression[38]. Runx3 also suppresses the high expression of T-bet and Zeb2, which normally occurs in effector T cells[39].

It is believed that iNKT cells express activation markers in the steady state and can immediately respond to foreign substances upon ligand stimulation. On the other hand, several recent reports have indicated that not all iNKT cells return to a steady state after activation[13,16,17]. However, the process has not been elucidated. In this study, we performed a single-cell transcriptome analysis of Klrg1[+] cells from DC/Gal-immunized WT and Zeb2-cKO mice. We found that the NKT1 lineage in Klrg1[+] iNKT cells comprised several clusters: C7, C11, C1, C9, and C10. The present results provide new evidence using a single-cell transcriptome analysis that Zeb2 is critical for the differentiation to C11 from C7 without undergoing apoptosis. In fact, when WT and Zeb2-deficient Klrg1[+] iNKT cells were compared in terms of apoptosis (Annexin-V) and proliferation (Ki-67) at day 7, Zeb2-deficient Klrg1[+] iNKT (i.e., Klrg1[+] Cx3cr1[−]GzmA[−] iNKT) cells were more prone to apoptosis at the differentiation stage. Thus, Zeb2 is critical for the establishment of Klrg1[+] long-lived effector iNKT cells. The role of Zeb2 in iNKT cells is similar in some respects to its role in CD8[+] T cells and NK cells, and dissimilar in others. Furthermore, Klrg1[+] iNKT cells were also negative for PD-1, CTLA4, and Lag3, and could be enhanced by secondary effects, indicating that Klrg1[+] iNKT cells are not exhausted iNKT cells, but Zeb2-dependent long-lived effector iNKT cells. Among innate lymphocytes, an induction of long-term effects of effector memory like innate lymphocytes may lead to strengthened immune surveillance against tumor cells.

## Methods

**Study design**. The main objective of this study was to elucidate the intrinsic mechanisms of long-lived effector iNKT cells, which express Klrg1 and have the capacity of memory response and anti-tumor effects, mediated via ligand-specific immunization. Our specific hypothesis in this regard was that these Klrg1[+] iNKT cells can be memorized during the priming phase in peripheral organs via selective mechanisms associated with activation, which is regulated by a transcription factor.

On the basis of screening candidate transcription factors specifically expressed in Klrg1[+] iNKT cells using RNA-seq, we identified Zeb2 as the putative transcription factor. To investigate the potential role of Zeb2 in iNKT cell development and activation, we generated CD4-Cre Zeb2[f/f] (Zeb2-cKO) mice that specifically lacked T cell Zeb2 activity. Using these Zeb2-cKO mice, following antigen priming, we performed multi-modal analysis of iNKT development in the thymus and the differentiation of NKT cells in the periphery, and compared the effects with those detected among littermates or WT mice. To assess the kinetics and memory function of long-lived NKT cells, we conducted thymectomy and single-cell RNAseq analysis. Thymectomy was performed to distinguish between iNKT cells newly recruited from the thymus and those differentiated in the periphery. To elucidate the heterogeneity and differentiation trajectory of the Klrg1[+] iNKT cell population, we performed scRNAseq of Klrg1[+] iNKT cells in the activated phase after immunization. The sample sizes in all experiments deemed sufficient for statistical analysis were determined based our previous studies and preliminary experiments.

**Mice**. Specific pathogen-free 6–8-week-old C57BL/6 female mice were purchased from Charles River, Japan. B6;129(Cg)-Zeb2 < tm1.1Yhi > (Zeb2 [flox/flox]) mice were purchased from RIKEN BRC with permission from Dr. Y. Higashi (Osaka University)[40]. CD4-Cre Tg mice were kindly provided by Dr. Nakayama (Chiba University, Japan). T-cell-specific Zeb2 KO mice were obtained by crossing Zeb2[flox/flox] mice with CD4-Cre Tg mice. Rag1 KO (B6.129S7-Rag1[tm1Mom]/J) mice were purchased from the Jackson Laboratory. All mouse strains were backcrossed with C57BL/6 mice for 12 generations. B6.129S6-Tbx21 <tm1Glm > /J

(T-bet KO) mice were purchased from the Jackson Laboratory. Most of the mice used herein were 6–12 week-old female mice. All mice were maintained under specific pathogen-free conditions at the RIKEN animal facility, and all procedures were performed in compliance with protocols approved by the Institutional Animal Care Committee at RIKEN (AEY2022-020).

**Thymectomy**. Mice at 6–8 weeks of age were anesthetized and shaven around the throat. A small incision was made at the base of the manubrium for suction cannula insertion. Both thymic lobes were removed separately. For sham surgery, mice were operated upon to make a small incision, although the thymus was not removed After more than two weeks of post-operative recovery, the mice were immunized with DC/Gal.

**Cell preparation**. The thymus and spleen were homogenized through a 70 μm cell strainer, and the erythrocytes were lysed using ACK lysing buffer (GIBCO), followed by two washes in RPMI1640 (SIGMA). For the isolation of lung and liver mononuclear cells (MNCs), the minced tissues were digested with collagenase D (Roche) and then layered on Percoll gradients (40/70%) (Amersham Pharmacia Biotec), followed by centrifugation for 20 min at $1840 \times g$. Bone marrow-derived dendritic cells (BM-derived DCs) were generated from BM precursors in culture with GM-CSF and pulsed α-galactosylceramide (α-GalCer) (200 ng/mL) for 48 h from day 6 and matured by LPS as previously described[3]. One million α-GalCer-pulsed mature BM-DCs (DC/Gal) were intravenously injected into one mouse.

**Flow cytometry**. The following monoclonal antibodies (mAbs) were purchased from BD Biosciences, BioLegend, Thermo Fisher Scientific, abcam, and Santa Cruz Biotechnology: anti-mouse-CD4 (GK1.5), -CD8 (53.67), -CD11a (M17/4), -CD11b (M1/70), -CD16/32 (93), CD19 (6D5), -CD24 (M1/19), -CD27 (LG.3A10), -CD43 (S7), -CD44 (IM7), -CD45 (30-F11), -CD49d (R1-2), -CD62L (MEL-14), -CD69 (H1.2F3), -CD103 (2E7), -CD122 (TM-b1), -CD127 (A7R34), -CD244.2 (eBIO244F4) -CD279 (29F.1A12), -CCR6 (29-2L17), -CXCR5 (2G8), -CXCR6 (SA051D1), - Cx3cr1 (SA011F11), -NK1.1 (PK136), -NKG2D (CX5), -Ly-6C (AL-21), -Klrg1 (2F1), -TCRβ (H57-597), -Glut1 (EPR3915), -granzyme A (3G8.5), -T-bet (eBio4B10), -Gata3 (TWAJ), -RORγt (B2D), -PLZF (9E12), -Eomes (Dan11mag), -TCF-1 (S33-966), -Blimp-1 (5E), -Ki-67 (SolA15), and streptavidin-FITC, -BV605, -BV650, BUV395. PE- or APC-conjugated CD1d-tetramers were purchased from MBL. Biotin-conjugated Annexin-V was purchased from BioLegend. Fixable Aqua Dead Cell Stain Kit (Invitrogen) and Fixable Violet Dead Cell Stain Kit (Invitrogen) were used to eliminate dead cells. Intracellular staining for transcription factors was performed using a eBioscience Foxp3 Staining Buffer Kit. Intracellular staining for cytokines was performed using a BD Cytofix/Cytoperm Kit. The concentrations of the antibodies used are presented in Supplementary Table 1. The LSRFortessa X-20 instrument and FlowJo software were used for analysis.

**Cell sorting**. Lung iNKT cells (Klrg1[+] or Klrg1[−] TCRβ[+]CD1d-tet[+]) from non-immunized or DC/Gal-immunized mice were sorted using a FACSAria sorter (BD Biosciences) and used for qPCR. C1 (2B4[−] CXCR6[+] TCRβ[+]CD1d-tet[+]) and C2 (2B4[+] CXCR6[+] TCRβ[+]CD1d-tet[+]) iNKT cells from the thymus of naïve WT mice were sorted for transfer experiments. Lung Klrg1[+] iNKT cells were sorted from DC/Gal-immunized CD4-Cre negative Zeb2[f/f] littermate mice or Zeb2-cKO mice on day7 for scRNA-seq.

**Quantitative PCR assay**. To evaluate gene expression by iNKT cells, we performed cDNA synthesis and pre-amplification using RNA extracted from FACS-sorted iNKT cells (Klrg1$^+$ or Klrg1$^-$ iNKT cells) without purifying RNA, using a CellsDirect One-Step qRT-PCR Kit (Invitrogen) with a mixture of pooled gene-specific primers (0.2 μM each). After 18 cycles of pre-amplification (each cycle: 95 °C for 30 s, 60 °C for 4 min), an aliquot was a template for quantitative PCR using the FastStart Universal Probe Master (Roche), a gene-specific forward and reverse primer pair, and the corresponding FAM-labeled hydrolysis probe (Universal Probe Library Set, Roche). Primers as follows: *Ifng* forward: 5′-ATCTGGAGG AACTGGCAAAA-3′, *Ifng* reverse: 5′-TTCAAGACTTCAAAG AGTCTGAGGTA-3′, *Zeb2* forward: 5′-CCAGAGGAAACAA GGATTTCAG-3′, *Zeb2* reverse: 5′-AGGCCTGACATGTAGT CTTGTG-3′, *Gzma* forward: 5′-GGCCATCTCTTGCTACTCT CC-3′, *Gzma* reverse: 5′-CGTGTCTCCTCCAATGATTCT-3′, *Klrg1* forward: 5′-GGCTTGAGGAACATTGATGG-3′, *Klrg1* reverse: 5′-TCAAGCTGTTGGTAAGAATCCTC-3′, *Hprt1* forward: 5′-TCCTCCTCAGACCGCTTTT-3′, *Hprt1* reverse: 5′-CC TGGTTCATCATCGCTAATC-3′; probes as follows: *Ifng* (Roche Diagnostics 04686942001), *Zeb2* (Roche Diagnostics 04688015001), *Gzma* (Roche Diagnostics 04685105001), *Klrg1* (Roche Diagnostics 04685091001), *Hprt11* (Roche Diagnostics 04692128001). Quantitative PCR was performed using ABI PRISM 7000 (Applied Biosystems).

In some experiments, we conducted alternative qPCR analysis. Total RNA was extracted from sorted cells using Trizol (Invitrogen,). The 1st strand cDNA was synthesized with PrimeScript II 1st strand cDNA Synthesis Kit (Takara). RT-qPCR was performed using TB Green Premix Ex Taq II (Takara,) and following manufacturer's standard protocols. Primers as follows: *Ifng* forward: 5′-ATCTGGAGGAACTGGCAAAA-3′, *Ifng* reverse: 5′-TTCAAGACTTCAAAGAGTCTGAGGTA-3′, *Il4* forward: 5′-CACTTGAGAGAGATCATCGGCA-3′, *Il4* reverse: 5′-CCGAAAGAGTCTCTGCAGCTC-3′, *Il17a* forward: 5′-CAGGGAGAGCTTCATCTGTGT-3′, *Il17a* reverse: 5′-GCTG AGCTTTGAGGGATGAT-3′, *Tbx21* forward: 5′-GCCCACAAG CCATTACAGGA-3′, *Tbx21* reverse: 5′-AAATGAAACTTCCT GGCGCATC-3′, *Zeb2* forward: 5′-CCAGAGGAAACAAGGAT TTCAG-3′, *Zeb2* reverse: 5′-AGGCCTGACATGTAGTCTTG TG-3, *Hprt1* forward: 5′-CCTCCTCAGACCGCTTTTT-3′, *Hprt1* reverse: 5′-AACCTGGTTCATCATCGCTAA-3′.

Gene expression was measured using the $\Delta\Delta C_T$ method, in which HPRT1 expression was an internal control.

**Next-generation sequencing (RNA-seq)**. We have deposited the RNA-seq data of lung iNKT cells from naïve or DC/Gal immunized mice in the Gene Expression Omnibus (GEO) database with accession code GSE128069.

Following quality control and trimming using trimgalore software (https://github.com/FelixKrueger/TrimGalore, version 0.6.7), the sequence reads were mapped and counted using Kallisto (https://pachterlab.github.io/kallisto/, version 0.48.0) to mm10 cDNA library provided by NCBI RefSeq. Expression changes and statistically significance of the difference were calculated using DESeq2 algorithm (https://github.com/mikelove/DESeq2, version 1.38.2). Transcription factors were defined as having "DNA binding", "nucleus" and "regulation of transcription" annotations provided by the Gene Ontology database.

**10x genomics library preparation**. The scRNA-seq was prepared using Chromium Next Gel Bead In-Emulsion (GEM) Single Cell 5′ Reagent Kits v2 (10x Genomics, San Francisco, CA, USA). Briefly, FACS-sorted cells were washed once with PBS containing

0.1% bovine serum albumin. Approximately 6000 cells per sample were loaded onto Next GEM chip K (10x Genomics) and partitioned into GEMs in a Chromium Controller (10x Genomics). Single-cell RNA libraries were prepared according to the Chromium Next GEM Single Cell 5′ Reagent Kits v2 User Guide.

**Sequencing**. scRNA libraries were sequenced using a HiSeq X sequencer (Illumina) at a minimum sequencing depth of 20,000 reads/cell using read lengths of 26 bp for read 1, 8 bp for the i7 index 8 bp for the i5 index, and 90 bp for read 2.

**scRNA-seq data processing**. Single cell transcriptome analysis was performed using the software 10 x Genomics cellranger (https://support.10xgenomics.com/single-cell-gene-expression/software/overview/welcome, version 6.1.2) following a standard protocol. mRNA sequences were mapped to mm 10 reference genome and de-multiplexed read counts were normalized, decomposed, and clustered using the Seurat toolkit (https://satijalab.org/seurat/, version 4.0.3). mRNA classification (nuclear/mitochondrial), marker gene detection, and other feature extraction of clustered cells were done using our in-house programs (see Resources).

**Trajectory analysis**. After removing dead, duplet, and non-NKT1 cells defined by gene expression pattern, NKT1 cells were re-decomposed using UMAP implemented in Seurat. Then, trajectories among NKT1 clusters were analyzed using pyslingshot (version 0.0.2, https://github.com/mossjacob/pyslingshot), a Python implementation of Slingshot (version 2.7.0, https://github.com/kstreet13/slingshot).

**Statistics and reproducibility**. The data are shown as the means ±SEM. The number used for each experiment is shown in figure legends. The raw data related to this study has been presented in Supplementary Data 1. Statistical analyses were performed using StatMate (Ver 5.01). Differences were analyzed using a two-tailed Student's *t* test or ANOVA Tukey–Kramer test. A two-tailed value of $P < 0.05$ was considered statistically significant.

**Reporting summary**. Further information on research design is available in the Nature Portfolio Reporting Summary linked to this article.

### Data availability

RNA-seq data that support the findings of this study have been deposited in the Gene Expression Omnibus (GEO) database with accession codes GSE128069[17]. The scRNA-seq data of iNKT cells have been deposited with links to BioProject accession number PRJNA1021694 in the DDBJ BioProject database. All source data underlying the graphs presented in the main figures are available in Supplementary Data 1.

### Code availability

The code for data cleaning and analysis is provided as part of the replication package. It is available at https://github.com/takaho/nktsc for review.

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

## Acknowledgements

We thank M. Kronenberg (La Jolla Institute for Immunology) for the critical reading of the manuscript. We also thank Masami Kawamura, Yuko Kurokochi and Jun Shinga for technical assistance and support in this study. This study was supported by Japan Society for the Promotion of Science (JSPS) KAKENHI grants 26460583 (S.F. and K.S.) and funding from the RIKEN (RIKEN President's Discretionary Fund) (S.F.).

## Author contributions

Conceptualization: S.F. and K.S. Methodology: T.I., K.S., T.W., T.E. and S.F., Investigation: T.I., K.S., H.A., M.S. and H.N. Visualization: T.I., K.S., T.W., T.E. and S.Y., Funding acquisition: S.F. Project administration: S.F., Supervision: S.F., Writing original draft & editing: K.S. and S.F., Writing and review: I.T.

## Competing interests

The authors declare no competing interests.
