## [Peer Review File · Communications Biology]

Reviewers' comments:

Reviewer #1 (Remarks to the Author):

This manuscript by Iyoda et al. identified the transcription factor Zeb2 plays an essential role in the differentiation of long-lived Klrg1+ invariant natural killer T cells upon α -GalCer-loaded DCs immunization, while being dispensable for the development of iNKT cells in the thymus and maintenance in the spleen at steady state. The authors showed that Zeb2 is required for the differentiation of Klrg1+Cx3cr1+GzmA+ iNKT cells and the maintenance of long-lived Klrg1+ iNKT cells independent of the thymus. The authors also found Zeb2-deficient Klrg1+ iNKT cells underwent apoptosis and the Zeb2-deficient mice fail to maintain the antigen specific secondary response. The data in this study are clear and well-presented, but I still have a few concerns about the conclusions and writing as detailed below:

1. The authors showed that after DC/Gal immunization, Zeb2-cKO mice show reduced numbers of total Klrg1+ iNKT cells, while the Klrg1+Cx3cr1+GzmA+ iNKT populations (most likely resemble iNKT1 subset) were mostly diminished. Could the Klrg1+ iNKT cells also contain iNKT2 or iNKT17 subsets? Is Zeb2 being required for the differentiation of all subsets, or show a selective role in iNKT1 subset? Actually, the authors' single-cell transcriptome analysis of Klrg1+ iNKT cells did identify C4 as an iNKT2 subset.
2. The authors showed that both T-bet KO mice and Zeb2-cKO mice show a defect in Klrg1+ iNKT cell differentiation, and the remaining Klrg1+ iNKT cells in T-bet KO mice lost Zeb2 expression. But does the remaining Klrg1+ iNKT cell in Zeb2-cKO mice show normal T-bet expression? At which stages of iNKT cell development are T-bet and Zeb2 being expressed? Current evidence cannot fully support the conclusion that "T-bet is essential for Zeb2 induction in activated iNKT cells".
3. The fact that Klrg1+ iNKT contains both 2B4+ and 2B4- populations itself cannot demonstrate C1 and C2 iNKT cell subsets could differentiate to Klrg1+ iNKT cells. This conclusion lacks experimental evidence.
4. One main discovery of this study is "Zeb2 is dispensable for development of iNKT cells in the thymus neither for their maintenance in the peripheral tissues in the steady state" and "Zeb2 plays an essential role in the differentiation into the Klrg1+ Cx3cr1+GzmA+ iNKT cell population". The authors performed extensive analysis of iNKT subsets in the thymus at steady state, analyzed iNKT cells in the spleen at steady state, and also performed detailed analysis of lung and spleen iNKT cells after DC/Gal treatment. One important piece of data is lacking here, could the authors also determine whether lung iNKT cells develop normally in Zeb2-cKO mice at steady state?

Minor points:

1. Page 6, line 6: should be DC/Gal/naïve?
2. Page 8, lines 15-16: ref 29 should be 30? 30 should be 31?
3. Page 10, line 1: Supplementary Figure 4 does not have a panel c?

Reviewer #2 (Remarks to the Author):

In this manuscript, Iyoda et al. examine the role of the transcription factor Zeb2 in generating a population of iNKT cells after stimulation with dendritic cells loaded with alpha-galactosylceramide that they deemed to represent a memory cell type. They show that Zeb2 is induced in a T-bet-dependent manner in Klrg1+ cells that also express Cx3CR1 and granzyme A and that in absence of Zeb2 this population of iNKT cells is largely decreased. They go on to show some experiments to support the idea that in absence of Zeb2, this particular population of iNKT cells undergoes apoptosis. The experiments are rather straightforward and generally well controlled with a few exceptions.

Comments: The gating strategy for the identification of iNKT cells could be strengthened as in some cases the tetramer+ cells blend with TCRb+ cells and are not separated. For example, Fig 7. I assume

this would not change the results that are considering the frequencies of sub-gated populations but this is a concern. Perhaps the authors could contrast the results of tetramer+ cells, with tetramer-cells. Is Zeb2 having a unique role in iNKT cells?

The single cell RNA-seq data should be re-analyzed and the description of the analysis steps expanded. From the manuscript narrative, it seems that cells with high mitochondrial reads content and doublets were removed after dimensionality reduction and clustering. This is counter to the usual way of analyzing these types of data. Also, it's not clear that the data are not overclustered. Finally, how was the supposedly developmental trajectory calculated? Packages such as Monocle or Slingshot should also be used to complement the current analysis. It is not clear from Fig 6d that the trajectory of development really follows a C7, 11, 1, 9 path.

It is also not clear why Zeb2 does not seem expressed in any of the described clusters.

1 **Response to Referees**

2

3 **Reviewers' comments:**

4 **Reviewer #1**

5 *1. The authors showed that after DC/Gal immunization, Zeb2-cKO mice show reduced*
6 *numbers of total KlrG1+ iNKT cells, while the KlrG1+Cx3cr1+GzMA+ iNKT populations (most*
7 *likely resemble iNKT1 subset) were mostly diminished. Could the KlrG1+ iNKT cells also*
8 *contain iNKT2 or iNKT17 subsets?*

9 --- Thank you for highlighting this important point. On the basis of the data presented in **Fig. 2**,
10 we established that a majority of the KlrG1⁺ iNKT cell population obtained from DC/Gal-
11 immunized mice were KlrG1⁺Cx3cr1⁺GzMA⁺ iNKT cells that are apparently derived from the
12 NKT1 subset. Given our findings that the numbers of KlrG1⁺Cx3Cr1⁺GzMA⁺ iNKT cells in
13 DC/Gal-injected-Zeb2-cKO mice were substantially diminished, we demonstrated that
14 KlrG1⁺Cx3Cr1⁺GzMA⁺ iNKT, as long-lived memory iNKT cells, are Zeb2-dependent. As the
15 reviewer points out, we detected a low frequency of KlrG1⁺ Cx3Cr1⁻ GzMA⁻ iNKT cells in the T-
16 bet KO (NKT1-deficient mice) and Zeb2-cKO mice one week after an administration of DC/Gal
17 (i.e., at early time point after a vaccination). This indicates that KlrG1⁺ Cx3Cr1⁺ GzMA⁺ iNKT
18 cells are derived only from the NKT1 subset, whereas KlrG1⁺ Cx3Cr1⁻ GzMA⁻ iNKT cells may be
19 derived from other NKT subsets, probably other types of NKT1, NKT2, and NKT17, and might
20 also include additional undetermined types of iNKT cells. Indeed, on the basis of our single-cell
21 analysis, we found that these cells also contained NKT2 subset-derived cells (cluster 4 in **Fig. 6**).
22 In addition, the population was also found to include an NKT1/NKT2 mixed type of NKT cells
23 in cluster 3 (**Fig. 6**). Regrettably, we were unable to determine the types of iNKT cells comprising
24 the mixed populations of NKT cells. Accordingly, as the reviewer indicates, KlrG1⁺ Cx3Cr1⁻
25 GzMA⁻ iNKT cells may be an interesting population, although on the basis of the current data, we
26 were unable to completely characterize all types of iNKT cells. To achieve a more comprehensive
27 characterization of these cells, we would need to analyze the NKT1, NKT2, or NKT17
28 subfamilies in greater depth by using NKT2- or NKT17-specific reporter mice in conjunction
29 with single-cell analysis, and this indeed should be a focus of our further studies in the near future
30 study.

31

32 *2. Is Zeb2 being required for the differentiation of all subsets, or show a selective role*
33 *in iNKT1 subset? Actually, the authors' single-cell transcriptome analysis of KlrG1+ iNKT cells*
34 *did identify C4 as an iNKT2 subset.*

35

36 ---It is known that Tbx21 is essential for the development of NKT1 in the thymus. However, the

37 development of NKT1, NKT2, and NKT17 in the thymus of Zeb2-cKO mice is normal (**Fig. 3**).
38 In response to the administration of DC/Gal, the Klr $g1^+$ Cx3Cr 1^+ GzmA $^+$ iNKT population, as
39 memory NKT cells, was reduced in the peripheral tissues of Zeb2-cKO mice. Given that we were
40 able to clearly detect and trace the fate of NKT1 from the activation to memory phase, we believe
41 that Zeb2 plays a role in differentiation from the activation of NKT1 cells to the maintenance of
42 long-term effector Klr $g1^+$ Cx3Cr 1^+ GzmA $^+$ iNKT cells in peripheral tissues (p. 10. line 195-197).
43 As mentioned in our response to comment 1, whereas we were unable to trace Klr $g1^+$ iNKT cells
44 derived from the other types of NKT subsets, given that NKT2 cells were clearly detected in the
45 C4 cluster in WT and Zeb2 cKO mice, we believe that the differentiation of NKT2 is not a Zeb-
46 dependent phenomenon (p. 11, line 230-233). On the basis of our findings in this study, we
47 conclude that Zeb2 plays a role in the differentiation of NKT from the activation of NKT1 cells
48 to the maintenance of long-term effector Klr $g1^+$ Cx3cr 1^+ GzmA $^+$ iNKT cells in peripheral tissues.

49

50 **3. The authors showed that both T-bet KO mice and Zeb2-cKO mice show a defect in**
51 **Klr $g1^+$ iNKT cell differentiation, and the remaining Klr $g1^+$ iNKT cells in T-bet KO mice lost**
52 **Zeb2 expression. But does the remaining Klr $g1^+$ iNKT cell in Zeb2-cKO mice show normal T-**
53 **bet expression?**

54 ---Thank you for raising this point. In this regard, we previously reported that in DC/Gal-
55 immunized mice, the expression of *Tbx21* in Klr $g1^+$ iNKT cells is considerably higher than that in
56 Klr $g1^-$ iNKT cells (Shimizu et al. PNAS 2014). In the present study, we compared the expression
57 of *Tbx21* in Klr $g1^+$ or Klr $g1^-$ iNKT cells in DC/Gal-injected Zeb2-cKO and WT mice (**Fig. 1g**) (p.
58 7, line124-123), and accordingly found that the expression of *Tbx21* in the Klr $g1^+$ iNKT cells of
59 DC/Gal-injected WT mice was higher than that in the same cells in DC/Gal-injected Zeb2-cKO
60 mice. In contrast, in DC/Gal-injected Zeb2-cKO mice, the expression of *Tbx21* in Klr $g1^+$ cells
61 was similar to the level of expression in Klr $g1^-$ iNKT cells of DC/Gal-injected WT and Zeb2-cKO
62 mice (**Fig.1g**). In summary, the frequency of NKT1-derived Klr $g1^+$ NKT cells was considerably
63 lower in Zeb2-cKO mice (Fig. 1e). In addition, the expression of T-bet in the remaining Klr $g1^+$
64 iNKT cells in these mice was lower than that of Klr $g1^+$ iNKT cells in DC/Gal-injected WT mice
65 (**Fig. 1g**) (p. 7). These observations thus provide evidence to indicate that the remaining Klr $g1^+$
66 iNKT cells in Zeb2-cKO mice contained much lower numbers of NKT1-derived cells.

67

68 **4. At which stages of iNKT cell development are T-bet and Zeb2 being expressed?**
69 **Current evidence cannot fully support the conclusion that “T-bet is essential for Zeb2 induction**
70 **in activated iNKT cells”.**

71 —Having sorted iNKT cells at each stage of the NKT cells in the thymus, we analyzed the
72 expression of *Tbx21* and *Zeb2* by qPCR (**Fig. 3f**) (p. 8, line 170-172). We found that T-bet was

73 characterized by a higher increase in expression during the later stages of iNKT cell development,
74 and established that Zeb2 is expressed at a lower level in the thymus. Nevertheless, the
75 development of NKT subsets was normal in Zeb2-cKO mice. Collectively, these finding provided
76 evidence to indicate that T-bet is essential for Zeb2 induction in activated iNKT cells.

77

78 **5. *The fact that Klr1+ iNKT contains both 2B4+ and 2B4- populations itself cannot***
79 ***demonstrate C1 and C2 iNKT cell subsets could differentiate to Klr1+ iNKT cells. This***
80 ***conclusion lacks experimental evidence.***

81 ---In accordance with this comment, we sorted C1 iNKT and C2 iNKT from the thymus of WT
82 mice, and each population was separately transferred to Rag-/- mice (**Supplementary Fig. 2f**).
83 On day 5 after transfer, we administered DC/Gal to these mice, and subsequently analyzed Klr1+
84 iNKT cells in the lungs 1 week later. We accordingly found that both C1 and C2 iNKT cell subsets
85 can undergo differentiation to Klr1+ iNKT cells. (**Supplementary Fig. 2g**) (p.9, line 176-182)

86

87 **6. *One main discovery of this study is “Zeb2 is dispensable for development of iNKT***
88 ***cells in the thymus neither for their maintenance in the peripheral tissues in the steady state”***
89 ***and “Zeb2 plays an essential role in the differentiation into the Klr1+ Cx3cr1+Gzma+ iNKT***
90 ***cell population”. The authors performed extensive analysis of iNKT subsets in the thymus at***
91 ***steady state, analyzed iNKT cells in the spleen at steady state, and also performed detailed***
92 ***analysis of lung and spleen iNKT cells after DC/Gal treatment. One important piece of data is***
93 ***lacking here, could the authors also determine whether lung iNKT cells develop normally in***
94 ***Zeb2-cKO mice at steady state?***

95 ---Thank you for this valuable suggestions. In the steady state, we found that in both Zeb2cKO
96 and WT mice, the cell number and cytokine expression of iNKT cells in the lungs were similar
97 with no statistically significant difference between the two groups being detected (p.6, line117-
98 119). In **Supplementary Fig.2b and 2c**, we have accordingly included these data of lung in the
99 steady state of WT and Zeb2cKO mice.

100

101 Minor points:

102 1. Page 6, line 6: should be DC/Gal/naïve?

103 -----Thank you for this comment. We have revised the text accordingly.

104 2. Page 8, lines 15-16: ref 29 should be 30? 30 should be 31?

105 -----Thank you for pointing out this issue. The citations were indeed incorrect, and we have
106 rectified as indicated.

107 3. Page 10, line 1: Supplementary Figure 4 does not have a panel c?

108 -----Thank you for indicating this error. We have revised accordingly.

109 **Reviewer #2**

110 *1. Comments: The gating strategy for the identification of iNKT cells could be*
111 *strengthened as in some cases the tetramer+ cells blend with TCRb+ cells and are not separated.*
112 *For example, Fig 7. I assume this would not change the results that are considering the*
113 *frequencies of sub-gated populations but this is a concern. Perhaps the authors could contrast*
114 *the results of tetramer+ cells, with tetramer- cells. Is Zeb2 having a unique role in iNKT cells?*

115 ---Thank you for raising these important points. As requested, we have added data for both
116 tetramer-negative and -positive cells (**Fig. 7a and 7b**). We confirmed that there were no
117 statistically significant difference between WT and Zeb2 cKO regarding the expression of
118 apoptotic and proliferation markers in tetramer-negative cells (p.13, line 263-265).

119

120 *2. The single cell RNA-seq data should be re-analyzed and the description of the analysis*
121 *steps expanded. From the manuscript narrative, it seems that cells with high mitochondrial*
122 *reads content and doublets were removed after dimensionality reduction and clustering. This*
123 *is counter to the usual way of analyzing these types of data. Also, it's not clear that the data are*
124 *not overclustered. Finally, how was the supposedly developmental trajectory calculated?*
125 *Packages such as Monocle or Slingshot should also be used to complement the current analysis.*
126 *It is not clear from Fig 6d that the trajectory of development really follows a C7, 11, 1, 9 path.*

127 ---Although the reviewer recommends the use of pseudo-time and trajectory analysis software
128 such as Monocle or Slingshot, these packages essentially implement exactly the same algorithm
129 as our method based on minimum spanning tree (MST) algorithm. Given that we sometimes use
130 Slingshot, permit us to explain in detail the differences between Slingshot and MST. Given that
131 Slingshot uses proprietary decomposition and clustering procedures, it often produces diagrams
132 that are incompatible with other analyses. Furthermore, in this study, we wished to determine the
133 fate of iNKT cells, for the analysis of which, Slingshot and Monocle are unsuitable. Conversely,
134 as indicated in our data, MST clearly reveals that as a cluster, apoptotic or dead cells are separated
135 from other clusters on scatterplot. For comparative purposes, below we provide a diagram
136 showing data generated using a Slingshot method. Notably, the pseudo-time map and the proposed
137 trajectory diagram show erroneous results, as the dead cells were found among the cells of some
138 other clusters. This is why we preferentially used our MST algorithm, based on trajectory analysis,
139 to separate the apoptotic or dead cells, rather than using Monocle or Slingshot.

140

141 Fig. 1. Trajectory and pseudo-time diagram obtained using the Slingshot package.

142

143 3. *It is also not clear why Zeb2 does not seem expressed in any of the described clusters.*

144 ---Thank you for providing us with an opportunity to clarify this point regarding the apparently
 145 low expression of Zeb2 in the clusters we described. Owing to the high threshold in the expression
 146 chart, it appears that limited population of cells in the cluster expressed Zeb2. By applying the
 147 upper 20% percentile as the threshold (please see the attached figure), we indeed found that many
 148 of the iNKT cells in these cluster express Zeb2 (**Supplementary Fig. 6**). Accordingly, given that
 149 the expression levels shown in the scatterplot represent relative expression rather than absolute
 150 gene expression, this may have led to some confusion. We have accordingly included this
 151 explanation in the revised manuscript (p.11-12, line 235-238).

152

REVIEWERS' COMMENTS:

Reviewer #1 (Remarks to the Author):

The authors have properly addressed all of my concerns.

Reviewer #2 (Remarks to the Author):

This is a review regarding the resubmission of the manuscript by Iyoda et al. The authors have addressed some of the previous comments but problems remain with regards of how the single cell RNA sequencing data have been analyzed. Duplicates and dead cells should be removed first and then the remaining cells should be analyzed and clusters identified. The expression of Zeb2 and KLGR1 in these data is displayed as binary due to the threshold applied as the upper percentile of 20%. This is not detrimental to the manuscript but is rather unusual.

Response to Reviewer #2

Reviewers' comments:

Reviewer #2

1.

This is a review regarding the resubmission of the manuscript by Iyoda et al. The authors have addressed some of the previous comments but problems remain with regards of how the single cell RNA sequencing data have been analyzed. Duplicates and dead cells should be removed first and then the remaining cells should be analyzed and clusters identified. The expression of Zeb2 and KLGR1 in these data is displayed as binary due to the threshold applied as the upper percentile of 20%. This is not detrimental to the manuscript but is rather unusual.

--- Figure 6 has been revised according to the reviewer's opinions. After removing dead and duplet cells, iNKT cells were re-decomposed using UMAP implemented in Seurat (new Fig. 6a) (at page 11). Then, trajectories among the NKT cell clusters were analyzed using psyingshot, a Python implementation of Slingshot (new Fig. 6d). We have described the methods in detail (p. 20). We have revised this part as demonstrated in new Figure 6a and 6d.

In addition, according to the expression of Zeb2 and Klrg1, we did not use a threshold, but $\log_2(\text{TPM}+1)$ in new supplementary Fig. 6. We have provided the relevant information in the results. (page 11).